# Hypoglycemic Activity of Self-Assembled Gellan Gum-Soybean Isolate Composite Hydrogel-Embedded Active Substance-Saponin

**DOI:** 10.3390/foods11223729

**Published:** 2022-11-20

**Authors:** Tao Wu, Jinghuan Cheng, Jinxuan Zhang, Hongxi Zhao, Wenjie Sui, Qiaomei Zhu, Yan Jin, Min Zhang

**Affiliations:** 1State Key Laboratory of Food Nutrition and Safety, Food Biotechnology Engineering Research Center of Ministry of Education, College of Food Science and Engineering, Tianjin University of Science and Technology, Tianjin 300457, China; 2China-Russia Agricultural Processing Joint Laboratory, Tianjin Agricultural University, Tianjin 300392, China

**Keywords:** gellan gum, soybean isolate protein, hydrogel, kidney tea saponin, encapsulation, hypoglycemic

## Abstract

In order to avoid hemolysis caused by direct dietary of kidney tea saponin, complex gels based on gellan gum (GG) and soybean isolate protein (SPI) loaded with saponin were created in the present study by using a self-assembly technique. Studies were conducted on the rheological characteristics, encapsulation effectiveness, molecular structure, microstructure, and hypoglycemic activity of GG/SPI-saponin gels. Increasing the concentration of SPI helped to enhance the strength and energy storage modulus (G′) of the gels, and the incorporation of high acylated saponin allowed the whole gel to undergo sol–gel interconversion. The encapsulation efficiency showed that GG/SPI-saponin was 84.52 ± 0.78% for saponin. Microstructural analysis results suggested that GG and SPI were bound by hydrogen bonds. The in vitro digestion results also indicated that saponin could be well retained in the stomach and subsequently released slowly in the intestine. In addition, the in vitro hypoglycemic activity results showed that the IC_50_ of encapsulated saponin against α-glucosidase and α-amylase were at 2.4790 mg/mL and 1.4317 mg/mL, respectively, and may be used to replace acarbose for hypoglycemia.

## 1. Introduction

A range of saponins with hypoglycemic effects can be found in many terrestrial plants, including Panax ginseng, gynostemma, yucca, maize mullein, smallpox, and other common herbs for the treatment of diabetes [1]. Kidney tea, also known as cat’s whiskers and buddleia, is a perennial herb of the genus Clerodendranthus in the family Lamiaceae. The chemical components that have been isolated include flavonoids, phenolic acids, and terpenoids, among which the terpenoids are mainly diterpenoids and triterpenoids, and triterpene saponins are the main class of saponins, consisting of 30 carbon skeletal triterpene glycosides [2].

Saponins have numerous physical properties including foaming and emulsification [3] and are used to make detergents, foaming agents, and emulsions. Their pharmacological effects namely anticancer and inhibition of cytotoxicity [4], antidiabetic activity [5], anti-inflammatory activity, and immunomodulatory activity [6]. Studies have shown that triterpene saponin has hypoglycemic effects [5], and kidney tea saponin possesses a triterpene saponin structure. The potential hypoglycemic activity of triterpene saponins is partly due to their ability to inhibit various enzymes in the small intestine, including α-glucosidase, protein tyrosine phosphate (PTP1B) [7], and α-amylase [8]. On the other hand, saponins can protect the pancreatic β-cell lineage from high glucose damage [9]. In diabetes, hyperglycemia generates reactive oxygen species (ROS), which in turn lead to lipid peroxidation and membrane damage, and these free radicals play an important role in the development of secondary complications of diabetes (renal, ocular, vascular, and neurological damage) [9]. Whereas saponins are hemolytic in nature, saponins can form complexes with sterols on the erythrocyte membrane, leading to increased permeability and subsequent loss of hemoglobin [10], and also inducing intestinal inflammation and impairing intestinal barrier function [11]. Therefore, the scope of its application is limited.

Gellan gum (GG) is a microbial extracellular anionic polysaccharide produced by the fermentation of Pseudomonas plantarum gynostemma [12]. It consists of four polysaccharides: (1,3)β-D-glucose, (1,4)β-D-glucuronic acid, (1,4)β-D-glucose, and (1,4)α-L-rhamnose [13], which undergo random curling at high temperatures and change to a double helix structure as the temperature decreases gel formation. Due to the presence of carboxyl groups in the glucuronide residues, the polymers have a net negative charge, which is critical to the ability of the GG to form hydrogels. In order to form strong and durable hydrogels, cations are used to make them form stronger gels [14]. The presence of cations during the sol–gel transition results in the formation of stronger gels. Monovalent cations form moderately strong hydrogels through electrostatic interactions with carboxylate groups, and the presence of divalent cations also inhibits the ability of unhydrated GG to hydrate [15]. Gels formed by monovalent cations of sodium or potassium ions can be recovered upon heating, while gels of magnesium or calcium salts cannot be recovered. Moreover, this polysaccharide produces gels that are transparent, acid-resistant, and heat-resistant [16], and is commonly used commercially as a food thickener and stabilizer, as well as a substitute for pectin in making jams and jellies. Microbial extracellular polysaccharides are becoming useful components of man-made products in chemical, pharmaceutical, cosmetic, and food factories [17], and have a wide range of commercial uses that are expected to be widely used in food and biotechnology.

Soybean isolate protein (SPI) is a resource-rich, inexpensive, high quality vegetable protein with high nutritional value, cholesterol-lowering, blood pressure-lowering [18], blood sugar-lowering [19] and good gelling and emulsifying effects [20], mainly composed of 7S and 11S [19]. SPI solidifies the protein matrix by de-folding, dissociating and forming aggregates through changes in hydrogen bonding, Van der Waals forces and hydrophobic interactions when heated. SPI hydrogels can encapsulate nutrients in the network structure [21] and are used for targeted delivery of nutrients due to their solubilization as well as release properties [22]. However, when SPI is used as a carrier of nutrients, this deficiency can be well avoided due to the fact that proteins are easily degraded by pepsin when they enter the human digestive tract and reach the stomach, thus releasing a large number of nutrients in combination with the crystalline gels [23]. Compared to synthetic polymers, protein-based hydrogels are popular in terms of amphiphilicity, biodegradability, biocompatibility [24], high nutritional value, excellent functional properties, and low toxicity [25].

Hydrogels have unique hydrophilic, biodegradable, permeable, and high swelling properties [26]. Currently, hydrogels are one of the most promising drug delivery carriers. Hydrogels have a rich three-dimensional network, which is beneficial for drug delivery [27]. In addition, hydrogels have swelling properties when exposed to water, and the swelling process is the drug release process, which can enhance the drug residence time [28].

In our study, we used SPI and GG to encapsulate kidney tea saponins, and it was expected that the hydrogels would form a tight three-dimensional network structure that would protect the saponins from degradation in the gastrointestinal tract and deliver them to the intestine. To test this hypothesis, we performed several experiments, such as in vitro digestion and observation of apparent morphology. Additionally, we performed in vitro hypoglycemic assays to evaluate the hypoglycemic activity of the composite hydrogels loaded with saponins.

## 2. Materials and Methods

### 2.1. Materials

High acyl gellan gum (99% purity) was produced by Shanghai Macklin Biochemical Co., Ltd. (Shanghai, China). SPI (90%, protein content) was purchased from Wanbang Industrial Co., Ltd. (Zhengzhou, China), and kidney tea saponin was produced by Xi’an Xinlu Biotechnology Co. (Xi’an, China). Pepsin, trypsin, and bile extract were obtained from Yuanye Biotechnology Co. (Shanghai, China). Potassium chloride and hydrochloric acid were purchased from Sail Chemical Technology Co., Ltd. (Tianjin, China). α-glucosidase, acarbose, PNPG, PNP, and DNS were purchased from Dingguo Biotechnology Co. (Shanghai, China).

### 2.2. Preparation of Saponin Hydrogels

The content of GG and SPI affects the viscosity of the hydrogel and the network structure of the hydrogel, too high viscosity will lead to uneven mixing of the saponin later and the desired sample cannot be formed successfully. GG was dissolved in deionized water so that the concentration of GG was 0.3% and keeping the same total volume of each sample, different concentrations of SPI (0%, 1%, 2%, 3%, 4%, 5%) were added to the deionized water and stirred magnetically overnight at room temperature, after which different concentrations of SPI were added to the GG solution, and the co-mix was heated at 80 °C for 1 h while KCl was added, removed and stored at 4 °C for 24 h to allow complete hydration to form GG-SPI composite gels. The ratios of gellan gum and soy protein isolate were 0.3:0 (S0), 0.3:1 (S1), 0.3:2 (S2), 0.3:3 (S3), 0.3:4 (S4) and 0.3:5 (S5), respectively.

### 2.3. Determination of Encapsulation Efficiency

The embedding rate of kidney tea saponin was determined by UV spectrophotometer, first the gel of embedding saponin was lyophilized, the powder was dissolved in distilled water, stirred overnight, centrifuged at 6000 rpm for 20 min, the supernatant was taken, diluted, and the absorbance value was measured at 243 nm with UV visible spectrophotometer, and the amount of saponin embedding was calculated according to the standard curve of saponin. The embedding rate of riboflavin was calculated by the following equation:(1)$$\mathrm{Embedding}\;\mathrm{rate}\;(\%)=\frac{C_{0}}{C_{1}}\;\times \;100$$

*C*_0_: saponin content embedded in the gel, mg; *C*_1_: total amount of saponin, mg.

### 2.4. GG-SPI Texture Analysis

The GG-SPI gels were made of the same height (40 mm) and diameter (50 mm) to ensure a flat surface and bottom, and the hardness of the gels was analyzed using a P/40 mm probe (Stable Micro Systems Ltd., Surrey, UK) with the following measurement parameters: pre-measurement speed of 8 mm/s, measurement speed of 2 mm/s, post-measurement speed of 10 mm/s, strain displacement of 70%, initiation force of 5 g, and the type of initiation was automatic.

### 2.5. Rheological Measurements

Rheological measurements of S0–S5 were investigated using a Haake Mars60 rheometer (Thermo Fisher Scientific, Dreieich, Germany). Parallel plates with a diameter of 20 mm and a gap of 1.0 mm were used for the measurements. First, oscillatory strain scan experiments were performed at 25, 50, and 75 °C over a strain amplitude range of 0.1–10% at a frequency of 1.0 Hz to determine the linear viscoelastic region, and a strain amplitude of 1% was selected for the following tests. Then, temperature scan tests were performed in the range of 75 to 25 °C with a cooling rate of 2 °C/min and a frequency of 1.0 Hz frequency scan experiments were performed from 0.1 to 10 Hz at 25 °C [29], and moisture evaporation was minimized by coating the periphery of the samples with light silicone oil during the measurements.

### 2.6. Fourier Transform Infrared Spectroscopy (FTIR)

The FTIR spectra of GG-SPI gels (iS50, Thermo Scientific, Waltham, MA, USA) were determined by the KBr press method. An accurate weighing of 1 mg of lyophilized sample of GG-SPI gel was added to 150 mg of fully dried potassium bromide (where potassium bromide was dried in an oven at 105 °C for more than 8 h), which was well ground with a mortar and pestle until finely ground against the wall. The mixed powder was pressed into a transparent sheet for about 30 s^–1^ min using a tablet press (about 10 × 10^7^ Pa), and the full-band scan (4000 cm^−1^~400 cm^−1^) was performed with an infrared spectrometer with a resolution of 4 cm^−1^ and 16 scans, and the sample infrared spectra were obtained by FTIR with air as the acquisition background.

### 2.7. Thermogravimetric Analysis (TGA)

The thermal stability of the gels was analyzed by TGA Q50, TA Instrument, USA. The GG-SPI gels were lyophilized by liquid nitrogen flash-freezing, and the samples were weighed about 3~5 mg and placed in an alumina crucible to start the program. The scanning temperature range was set to 30~600°C, the heating rate was 10 °C/min, and the sample was warmed up under a flowing nitrogen atmosphere at a nitrogen flow rate of 25 mL/min. The instrument recorded the change curve of sample mass with time and temperature during the warming up process.

### 2.8. Scanning Electron Microscopy (SEM)

The GG-SPI were lyophilized by liquid nitrogen, and the natural lyophilized sections were taken and placed under a scanning electron microscope (JSM-IT300, JEOL Ltd., Tokyo, Japan) after surface spraying with gold to observe the microscopic morphology of the composite gels.

### 2.9. Confocal Laser Scanning Microscope (CLSM)

The distribution of GG and SPI was observed using a laser scanning confocal microscope (Zeiss LSM 980, Jena, Germany). Gellan gum was covalently labeled with FITC, specifically, gellan gum and 50 mL of deionized water were mixed, then 10 μL/mL of FITC was added to the mixture, the reaction mixture was protected from light, stirred overnight, and then fully dialyzed against distilled water in the dark for 12 h before freeze-drying. Nile blue was used for non-covalent labeling of soybean isolated proteins. An amount of 50 mL of protein suspension was prepared first, and then 10 μL/mL Nile blue was added to the protein suspension. The mixture was stirred at room temperature overnight, then dialyzed in distilled water for 12 h in the dark and freeze-dried. The labeled proteins were dissolved in water and stirred overnight at room temperature and labeled gellan gum was added in different concentrations of SPI solution (0%, 1%, 2%, 3%, 4%, 5%, *w/v*) to make the concentration of GG (0.3%, *w/v*) and keep the same total volume of each sample, and 0.1% KCl was added to the solution and mixed well. The samples were placed on microscope slides, covered with a thin film, and fluorescence images were obtained simultaneously at 488 nm and 633 nm, and the above experiments were protected from light [30].

### 2.10. In Vitro Gastrointestinal Simulation

To investigate the release properties of saponin, in vitro simulated digestion experiments were performed, and in vitro gastrointestinal simulations were performed with reference to previously published methods [29]. Preparation of simulated gastric fluid: 2.0 g of NaCl was added to 1 L of deionized water, and pepsin (4000 U/mL) was added after adjusting the final pH of the digestion solution to pH 2.0 with HCl (0.1 M). Preparation of simulated intestinal solution: 6.8 g of dipotassium hydrogen phosphate was added to 500 mL of deionized water, the digest was adjusted to pH 7.5 with sodium hydroxide (0.1 M), and then diluted to 1 L with deionized water, trypsin (100 U/mL) and bile extract were added.

An amount of 1 g of sample was used for digestion experiments. Digestion was carried out at 100 rpm and 37 °C. After 120 min of simulated gastric digestion, intestinal digestion was added for 360 min. After simulating gastric digestion for 120 min, intestinal digestion was added and digested under the same conditions for 360 min. Every half hour, a portion of the digest was removed and centrifuged at 5000 rpm for 10 min, and the supernatant was collected for the determination of saponin concentration. The content of saponins was determined.

### 2.11. In Vitro Hypoglycemic Assays

Using pNPG as the substrate and acarbose as the positive control, briefly, 20 μL of 0.22 U/mL α-glucosidase solution, 140 μL of phosphate buffer solution, and 20 μL of different concentrations of sample solution were added to a 96-well plate and incubated at 37 °C for 20 min, followed by 50 μL of pNPG solution (10 mmol/L) and incubated at 37 °C for 20 min. Finally, 80 μL of 0.4 mol/L sodium carbonate solution was added to terminate the reaction. The sample absorbance was measured at 405 nm, and the sample control was replaced by buffer solution instead of α-glucosidase solution. The enzyme solution was replaced by buffer solution. The IC_50_ could be calculated based on the inhibition rate of the samples at different concentrations using acarbose as a positive control [8]. The inhibition rate was calculated as:(2)$$\mathrm{Inhibition}\;\mathrm{rate}\;(\%)=1-\frac{A_{\mathrm{sample}}{\;-\;A}_{\mathrm{sample}\;\mathrm{control}}}{A_{\mathrm{control}}}$$

Mix 10 μL α-amylase solution (0.5 U/mL) with 10 μL sample solution (dissolved in DMSO), incubate at 37 °C for 15 min, then add 500 μL 0.8% soluble starch solution (dissolved in 1 mmol/L pH 6.8 potassium phosphate buffer), incubate at 37 °C for 15 min, finally, add 500 μL DNS reagent. The reaction was terminated by adding 500 μL of DNS reagent in a boiling water bath for 10 min and then diluted with 4 mL of distilled water, 240 μL was transferred to a 96-well plate and the absorbance was measured at 540 nm. The sample control group used buffer solution instead of α-amylase solution; the blank group used DMSO instead of sample solution; the blank control group used buffer solution instead of α-amylase solution and DMSO instead of sample solution; acarbose was used as positive control. The IC_50_ can be calculated based on the inhibition rate of the samples at different concentrations [31]. The inhibition rate is calculated as:(3)$$\mathrm{Inhibition}\;\mathrm{rate}\;(\%)=1-\frac{A_{\mathrm{sample}}{\;-\;A}_{\mathrm{sample}\;\mathrm{control}}}{A_{\mathrm{control}}{\;-\;A}_{\mathrm{blank}\;\mathrm{control}}}$$

### 2.12. Statistic Analysis

All experiments were repeated three times. The results of our experiments were described and analyzed using Origin software (version 8.0, Minneapolis, MN, USA). Statistics were produced by GraphPad Prism 8 (GraphPad Software, San Diego, CA, USA). Results were analyzed using one-way ANOVA (analysis of variance). Significance was tested using Dunnett’s multiple comparison test. Significant differences were found between samples with different letters (*p* < 0.05).

## 3. Results and Discussions

### 3.1. Encapsulation Efficiency

Encapsulation efficiency (EE) is an important characteristic of hydrogels, as shown in Table 1, except for S0 (blank), the encapsulation rate of composite gels increased gradually with increasing SPI concentration, and S4 reached a maximum of 84.52 ± 0.78% at SPI concentration of 3%. For instance, the encapsulation rates of vitamin C (VC) in liposomes and chitosan nanoparticles were 48.30% [32] and 15.70% [33], respectively. In contrast, the self-assembled hydrogel formed by bovine serum albumin-citrus peel pectin reached 65.3% of VC encapsulation [34]. Spray drying of maltose dextrin resulted in an EE of 73.81% for steviol glycosides, while electrospraying with corn alcohol soluble protein increased the EE to 75.87% [35]. The addition of SPI significantly increased the viscosity of the hydrogel system, and with the increase in SPI content, the embedding rate showed a trend of first increasing and then decreasing, indicating that the viscosity of the composite gel system was too large and also unfavorable for the embedding of saponins, and S4 was considered to develop the most suitable experimental conditions for the development of thermally stable hydrogels of gellan gum and soybean isolate protein.

### 3.2. Texture Analysis of Gel

The hydrogel gel strength was influenced by the soybean isolate protein content. As shown in Figure 1, the addition of soybean isolate protein to the gellan gum can improve the gel strength of the composite hydrogel, and the strength of the hydrogel increases with the increase in SPI content. The addition of a large amount can lead to excessive viscosity of the hydrogel thermal solution making it impossible to mix the encapsulated material uniformly, resulting in a decrease in hardness [36]. When the amount of soybean isolate protein exceeds the optimal dose, the gel strength of the hydrogel decreases with the increase in protein content, which may be due to the collapse of the three-dimensional network of the hydrogel caused by a large amount of protein filling, making the hardness decrease. S3 composite hydrogel has the highest gel strength, while the gel strength of S4 and S5 composite hydrogels gradually decreased.

### 3.3. Rheological Analysis

The dynamic rheological properties of the GG/SPI composite system can reflect the viscoelasticity of the composite system. The energy storage modulus G′ reflects the elastic nature of the stored energy in the system and can recover its original shape and is related to the strength of the junction zone, the number of bonding, and the effective chains forming the network structure [37]; G″ can represent the viscosity of the sample, which is influenced by the friction, motion, and mobility of small molecules [38]. As shown in Figure 2, the energy storage moduli G′ and G″ of the hydrogels gradually increased with the increase in soybean isolate protein content, except for S0. The junction zone expanded with increasing biopolymer concentration, indicating that the increase in soybean isolate protein affects the rheological properties of the composite carrier structure. It can also be seen from the figure that the energy storage modulus (G′) of all hydrogels is higher than their loss modulus (G″) regardless of the SPI concentration, indicating the formation of GG/SPI hydrogels.

As shown in Figure 3, the G′ and G″ curves of S0 show an increasing trend after the G′–G″ intersection at 67.15 °C, which is the gel–sol transition temperature (Tsg) of S0. This is mainly due to the fact that the gellan gum shows a double helical structure at low temperatures and undergoes irregular curling with the increase in temperature. The temperatures at which G′ intersects G″ for S1–S5 are at 67.43 °C, 64.98 °C, 58.62 °C, 58.89 °C, and 74.65 °C, respectively. In the curves of G′ or G″ versus temperature for S0–S5, the energy storage modulus G′ gradually decreases with increasing temperature, intersects with G″ at the corresponding temperature, and G′ is greater than G″ before the intersection, which indicates that a gel is formed at low temperature, while with the increase in temperature, G′ is smaller than G″ and a sol is formed, realizing the gel–sol transition. The increase in temperature led to a decrease in G′ of the soybean isolate protein suspension, which could be attributed to the loss of stiffness due to the reduction of hydrogen bonds and electrostatic interactions in the protein molecules as a result of the initial heating [39]. When the temperature was further increased, the SPI molecules exposed more reactive groups and formed a sparse gel structure. In conclusion, the rheological properties of GG/SPI gels can be controlled by varying the concentration of SPI.

### 3.4. Structural Characteristics of Composite Gel

#### 3.4.1. Fourier Transform Infrared Spectroscopy (FTIR)

The structural characteristics and thermal stability of S4 were investigated by infrared spectroscopy and thermogravimetry, respectively, and we used infrared spectroscopy to understand the interactions between gellan gum, soybean isolate, and kidney tea saponin. As shown in Figure 4A, the main characteristic absorption peaks of SPI are the C=O stretching vibration absorption peak in the amide I band at 1672.84 cm^−1^, the N-H bending vibration absorption peak in the amide II band at 1523.83 cm^−1^, and the absorption peak at 1238.15 cm^−1^ caused by the C-H stretching vibration and N-H bending vibration in the amide III band. The C-H bending vibration peak at 1450.72 cm^−1^, the absorption peak at 1390.10 cm^−1^ due to asymmetric deformation of CH_3_, the C-O stretching vibration peak at 1076.46 cm^−1^, and the broader peak at 3286.22 cm^−1^ for the N-H stretching vibration of the protein [40]. The absorption bands of the GG were 3415, 2927, 1635, and 1418 cm^−1^, corresponding to the O-H stretching vibration, the C-H stretching vibration, the C-O stretching vibration of the carboxylic acid anion, and the C-H stretching vibration of the methyl group [41]. The difference between GG/SPI-saponin and saponin indicates that the O-H absorption band shifts from 3432 cm^−1^ to 3395 cm^−1^. Hydrogen bond formation reduces the density of bonding electron clouds, chemical bonding force constants, and stretching vibrational absorption shifts to lower wave numbers [42]. Thus, the findings also suggest that the main interaction between the gellan gum and soybean isolate is hydrogen bonding.

#### 3.4.2. Thermogravimetric Analysis (TGA)

Thermogravimetric analysis was performed to study the mass loss and degradation temperature of GG, SPI, saponin, GG-SPI, and GG/SPI-saponin to further determine their thermal stability. The lower sample mass loss rate reflects the higher stability of the samples at the same temperature. The thermal degradation of the hydrogels was determined over the temperature range of 30–500 °C. As shown in Figure 4B, thermal degradation of the pure saponins occurred first at 110 °C, where a mass loss was 3.99% and a mass loss of 2.26% for the temperature range of 110–175.93 °C were mainly caused by the physical and chemical evaporation of water, respectively [43]. Degradation at 175.93–500 °C is the breakdown of glycosidic bonds and intramolecular hydrogen bonds. The mass loss of GG/SPI-saponin distributed between 30–101.34 °C (6.68% mass loss), 101.34–196.08 °C (2.27% mass loss), 196.08–500 °C (59.39% mass loss). The causes of pyrolysis are evaporation of water vapor, breaking of side chains, and degradation of the main chain [44]. The maximum degradation temperatures of GG/SPI-saponin composite gels were significantly higher compared to pure saponin, indicating improved thermal stability of the saponin.

### 3.5. Scanning Electron Microscope (SEM)

The surface morphology of the gel was observed by scanning electron microscopy, and the gellan gum undergoes irregular curling during heating, while it undergoes helical upon cooling [45].

Figure 5A–F shows the images of S0–S5 at 100× magnification. The SEM images of GG/SPI composite hydrogels show that the pure gellan gum hydrogel (S0) had an inhomogeneous and rough structure with a lamellar aggregate honeycomb-like 3D network structure, and the pores of S0 were significantly larger than those of the other hydrogels. The addition of soybean isolate reduced the pore size and increased the density of the hydrogels. The gel structure becomes more compact and ordered with very small pores, indicating a higher crosslinking density of the polymer molecules [46]. The S1 and S2 composite hydrogels exhibit some density, while the pores of the composite gels gradually become neat with increasing SPI concentration, and SPI dominates the GG/SPI gel structure. At S4, the pores are uniformly dense and homogeneous. When its concentration is increased again, the pores of the hydrogel become larger and looser, which is consistent with encapsulation efficiency study mentioned previously. This also indicates that the prepared composite hydrogels can regulate their pore size by controlling the concentration of SPI and have great potential for the encapsulation and delivery of water-soluble nutrients.

### 3.6. Confocal Laser Scanning Microscope (CLSM)

The distribution of polysaccharides in the GG-SPI gel in the protein was observed by laser confocal microscopy, and the results are shown in Figure 6. Green is GG labeled with FITC and red is SPI labeled with Nile blue, and GG is more uniformly distributed in SPI [41]. The overlapping yellow region consisting of the SPI region (red) and GG region (green) indicates the formation of an interconnected structure between these two substances. As can be seen from the figure, the SPI concentration gradually increases and the yellow region shows a trend of increasing and then decreasing, reaching a maximum at S4, indicating that the two substances bind best at this concentration. This may be due to the heating of soybean isolate protein 7S globulin denaturation so that its hydrophilic group attached to the surface, hydrophilic ability to enhance the hydrogen bonding and charge interaction between soybean isolate protein and the anionic gellan gum, so that the two are better combined, which is also consistent with the results of infrared spectroscopy. Yet again, the increase in the content of the gellan gum resulted in the inability to fuse better, indicating that the binding between the two had reached saturation [47]. Because of the increased SPI content, larger irregular spherical aggregates appear, and the formed spherical aggregates may play a role in enhancing the gel hardness of the composite gels.

### 3.7. In Vitro Release Mechanism

#### 3.7.1. In Vitro Simulation Digestion

By measuring the release rate of encapsulants in gels under in vitro simulated conditions, the release performance of the composite gel system can be more directly understood, and it is also beneficial for the subsequent investigation of the release mechanism of core materials in the system [48].

As shown in Figure 7, the GG/SPI-saponin composite gel was released at a relatively low percentage during the gastric digestion phase. Only about 9.438% of saponin was released after 2 h of incubation in SGF. It is believed that this phenomenon is due to the rapid dehydration and shrinkage of the protein/polysaccharide composite gel after the addition of polysaccharide in acidic conditions in the stomach, which wraps around saponin and hinders the erosion of the protein by pepsin. When it was transferred to SIF and incubated for 4 h, the release of saponin reached 63.73%, and finally, in SCF, the final release of saponin reached 77.74%. Here, the large release of saponin in the intestine was mainly due to ion exchange, and in the intestinal environment, the pH shifted to neutral leading to massive water absorption and swelling of the gel and an increase in pore size, which allowed the rapid release of saponin.

#### 3.7.2. Release Model

In order to further investigate the saponin release mechanism of the composite gel, and after preliminary core release curve fitting, the Zero order and Korsmeyer–Peppas models are used in this paper to explore the release mechanism of cores in proteoglycan composite gels, and the Korsmeyer–Peppas release model has a high correlation coefficient [49]. So, they are used to fit the saponin release data of the composite gel with the following equation:(4)$$\mathrm{Zero-order}:\;\frac{M_{t}}{M_{\infty }}\;=\;100(1\;-\;k_{0}t)$$
(5)$$\mathrm{Korsmeyer-Peppas}:\;\frac{M_{t}}{M_{\infty }}\;=\;kt^{n}$$
where *M*_t_/*M*_∞_ is the proportion of active substance released at moment t, k is a constant incorporating these properties, and n gives an indication of the release mechanism. n is also the diffusion index, and the release mechanism of the active substance in the film, sheet, spherical, and column models can be inferred from the magnitude of the n value. Among them, for the columnar model, Fick diffusion is observed when n < 0.45; irregular diffusion (dissolution and concentration diffusion simultaneously) is observed when 0.45 < n < 0.89; and dissolution transfer diffusion is observed when n > 0.89. The correlation coefficient (r^2^) is a linear relationship between saponin release and time [49].

As can be seen from Table 2, the mode of release of active substances in composite gels mainly depends on the dissociation process of the gel network. For GG/SPI composite gels, the diffusion index n was greater than 0.89 under the Peppas model, indicating that the mode of saponin release was mainly solubilization transfer diffusion. The reasons for this diffusion are firstly, the digestive hydrolysis of SPI by pepsin and pancreatic enzymes, which disrupts the gel structure, and secondly, the electrostatic repulsion between SPI and GG, which are both negatively charged in the intestinal fluid (pH 7.0) [50], and the network structure of the gel tends to swell and loosen. The release of saponins was promoted due to the swelling of the gel and the digestive hydrolysis of the protein.

### 3.8. In Vitro Hypoglycemic Activity

To further explore the hypoglycemic activity of the encapsulated saponins, the in vitro α-amylase and α-glucosidase inhibitory activities of the positive control (acarbose), pure kidney tea saponin monomer and GG/SPI-saponin composite gel were investigated in this study. Table 3 shows the regression equations for the inhibition rates and semi-inhibitory concentrations of the three substances on α-amylase and α-glucosidase. R^2^ for each regression equation was 0.99, indicating a good fit. Acarbose was the most effective inhibitor of α-amylase, followed by pure saponin monomer and GG/SPI-saponin complex gel with semi-inhibitory concentrations of 0.4974, 1.4995, and 1.4317 mg/mL, respectively, and similarly, acarbose was the most effective inhibitor of α-glucosidase, followed by pure saponin monomer and GG/SPI-saponin complex gel with semi-inhibitory concentration values of 0.9293, 2.0565 and 2.4790 mg/mL, respectively.

Acarbose is a well-known and typical antidiabetic drug [51]. In contrast, the semi-inhibitory concentrations of encapsulated saponins on both α-glucosidase and α-amylase were not significantly different from those of pure saponin monomers. However, due to the hemolytic properties of pure saponins and the fact that conventional oral hypoglycemic agents for diabetes are also associated with adverse side effects such as hepatotoxicity, renal and kidney damage, and adverse gastrointestinal symptoms [52], encapsulated saponins may be safer.

Notably, previous reports suggest that mild α-amylase inhibition activity helps to alleviate gastrointestinal discomfort, as excessive inhibition of α-amylase may lead to abnormal fermentation of undigested starch bacteria in the colon [53]. This suggests that GG/SPI-saponin complex gel may also be useful for patients with type 2 diabetes [54].

## 4. Conclusions

The increased concentration of SPI allows the GG/SPI-saponin composite gel to possess stronger rheological properties compared to GG alone, which could improve the competitiveness of gellan gum as a gelling material. The change in absorption peaks in FTIR spectral analysis also confirmed the GG-SPI interaction, indicating that inter- and intra-molecular interactions of GG and SPI promoted the formation of the composite gel. The microstructure of the composite gel can be manipulated by changing the concentration of SPI. The increase in biopolymer concentration due to the increased level of cross-linking resulted in a dense gel structure and homogeneous pores. The improved gel structure resulted in better encapsulation efficiency of GG/SPI-saponin composite gels, which positively affected the loading effect of the gels. In addition, the GG/SPI composite gel protected the stability of saponin during gastric digestion and improved the release ratio of saponin during digestion in the small intestine. Notably, the encapsulated saponins still showed excellent hypoglycemic activity in in vitro hypoglycemic assays. Therefore, in our study, food-grade proteins and natural microbial polysaccharides were used to develop gellan gum/soybean isolate protein composite hydrogels, as an effective and safe method to protect saponin from degradation in the stomach and deliver them to the intestine, and GG/SPI as a potential delivery system with promising applications to improve the stability and bioavailability of functional agents.

## Figures and Tables

**Figure 1 foods-11-03729-f001:**
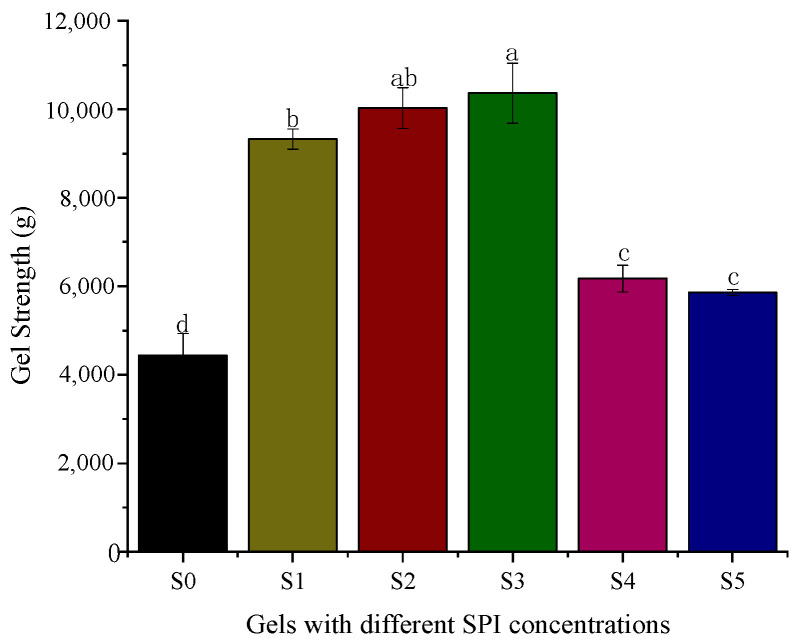
Gel strength of hydrogels. The ratios of gellan gum and soy protein isolate: S0 (0.3:0); S1 (0.3:1); S2 (0.3:2); S3 (0.3:3); S4 (0.3:4); S5 (0.3:5). There were significant differences among the samples with different letters (*p* < 0.05).

**Figure 2 foods-11-03729-f002:**
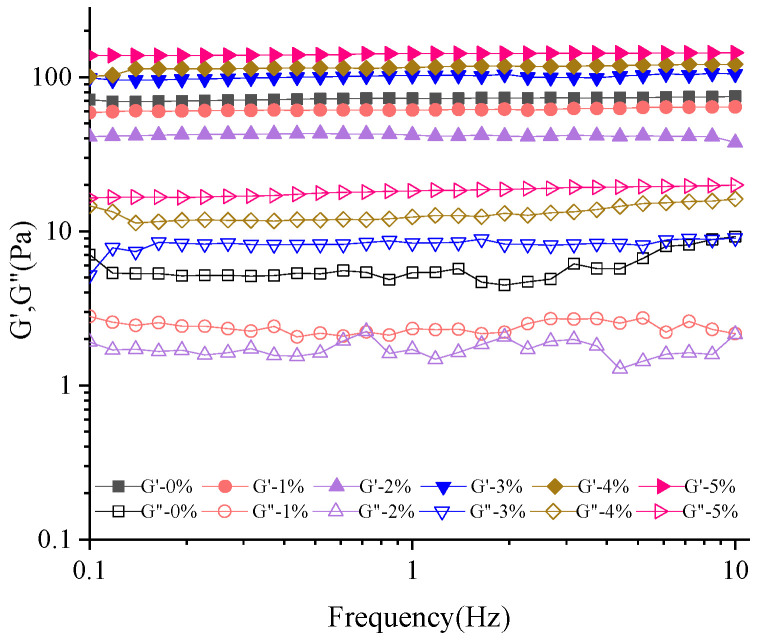
Storage modulus (G′) and loss modulus (G″) versus frequency for soybean isolate protein content in composite gels at 0–5%.

**Figure 3 foods-11-03729-f003:**
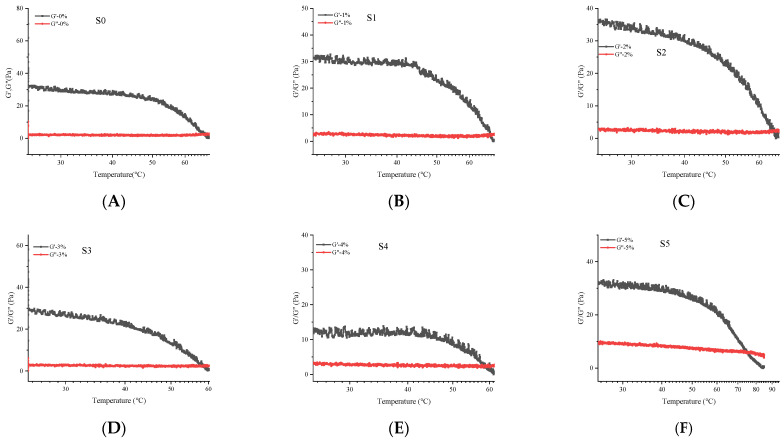
The storage modulus (G′) and loss modulus (G′′) of S0–S5 as a function of temperature at the frequency of 1 Hz. (**A**) S0; (**B**) S1; (**C**) S2; (**D**) S3; (**E**) S4; (**F**) S5. The ratios of gellan gum and soybean isolate were S0 (0.3:0); S1 (0.3:1); S2 (0.3:2); S3 (0.3:3); S4 (0.3:4); S5 (0.3:5).

**Figure 4 foods-11-03729-f004:**
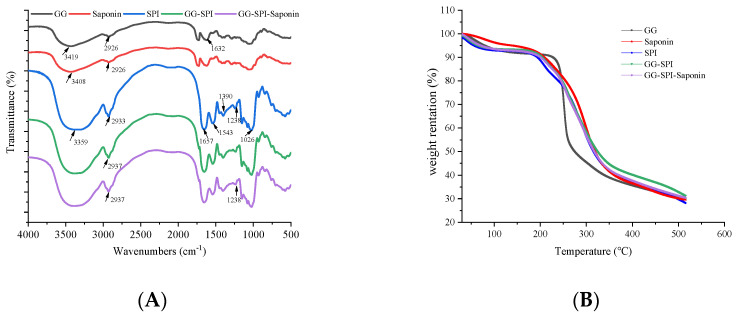
Structural characteristics of composite gel. (**A**) FT-IR spectra of GG, SPI, saponin, GG-SPI, and GG/SPI-saponin; (**B**) TGA thermograms of GG, SPI, saponin, GG-SPI, and GG/SPI-saponin.

**Figure 5 foods-11-03729-f005:**
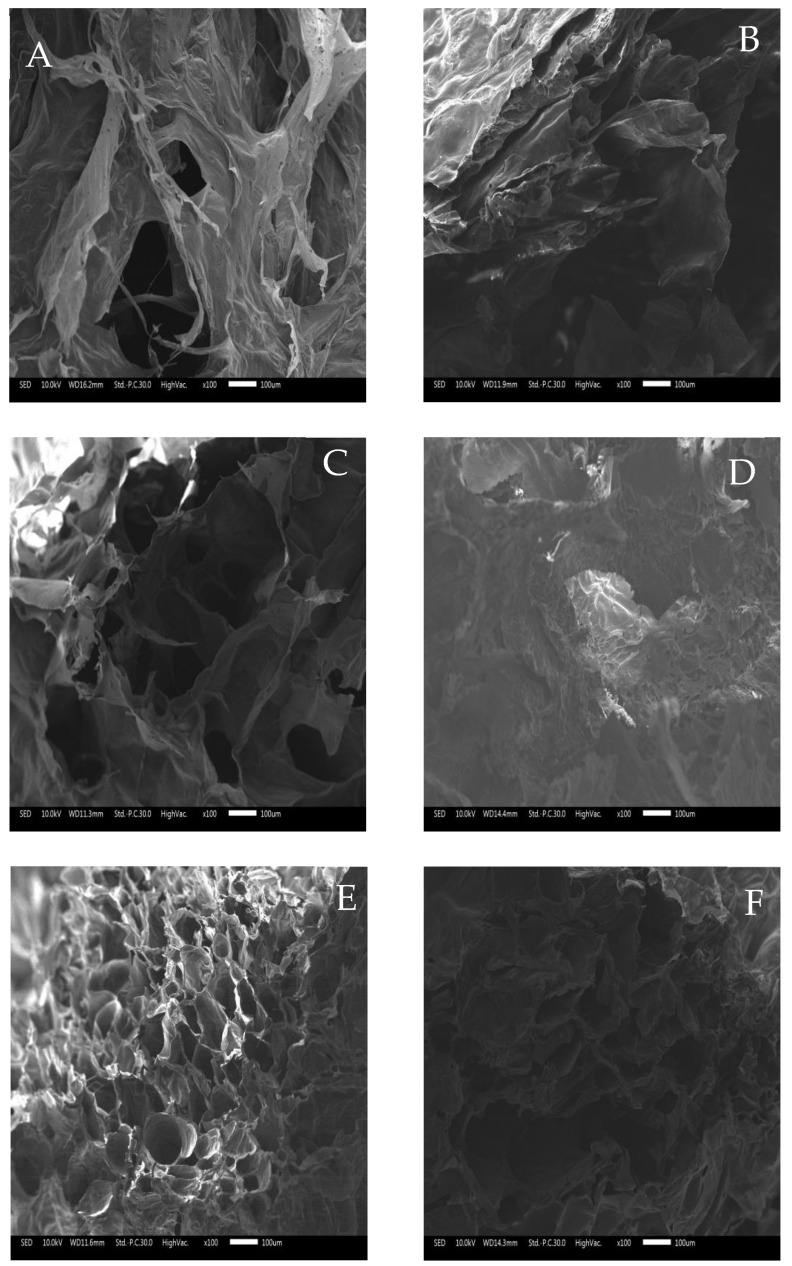
Morphology of hydrogels. (**A**) S0; (**B**) S1; (**C**) S2; (**D**) S3; (**E**) S4; (**F**) S5. The scale bars correspond to 100 μm. The ratios of gellan gum and soybean isolate were S0 (0.3:0); S1 (0.3:1); S2 (0.3:2); S3 (0.3:3); S4 (0.3:4); S5 (0.3:5).

**Figure 6 foods-11-03729-f006:**
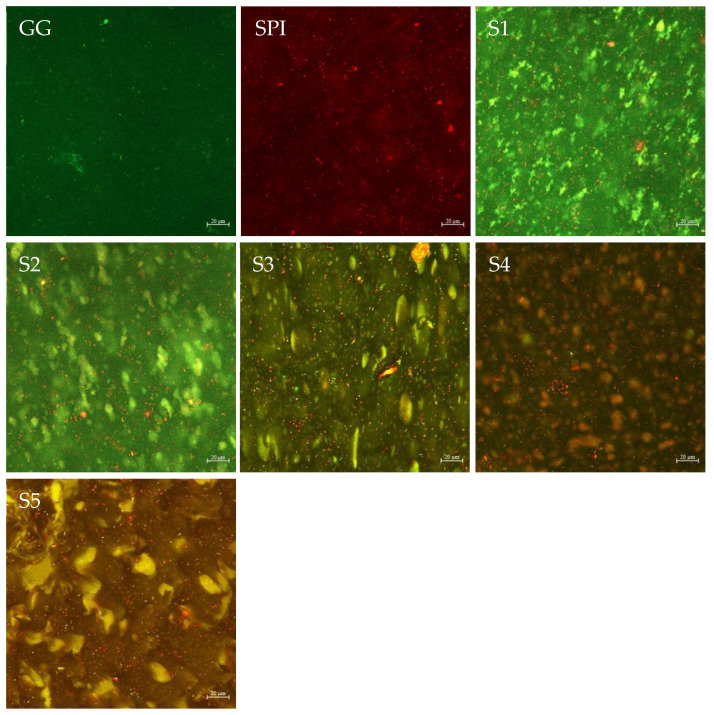
Effect of different SPI content on laser confocal microscopy of GG-SPI complex gel. The scale bars correspond to 20 μm.

**Figure 7 foods-11-03729-f007:**
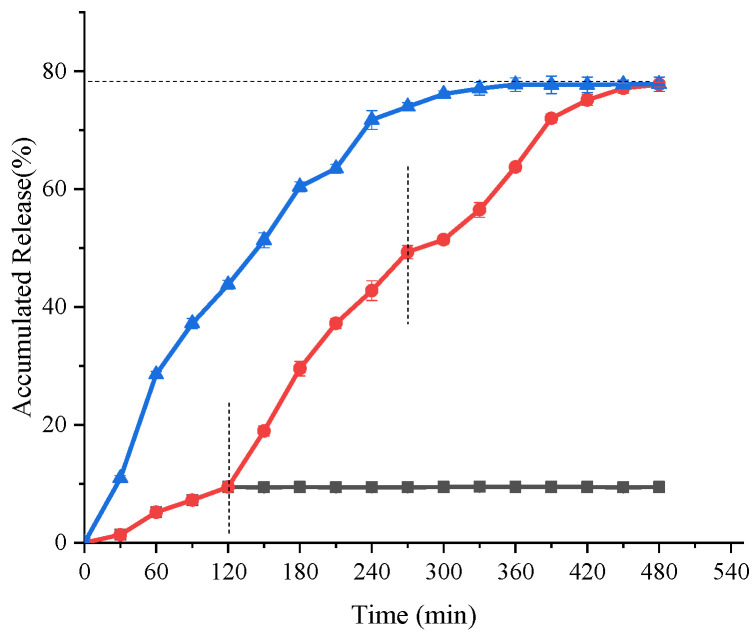
Releasing properties of saponin in vitro. Black curve shows the variation of saponin release from the composite hydrogel loaded with saponin in simulated gastric fluid (pH 2.0) as a function of time; red curve shows the cumulative release curve for the composite hydrogel samples loaded with saponin; blue curve shows the release of saponin in simulated intestinal fluid (pH 7.5) for the composite hydrogel samples loaded with saponin with time.

**Table 1 foods-11-03729-t001:** Encapsulation efficiency of hydrogels.

Name of Hydrogel	Encapsulation Efficiency (%)
S1 ^1^	60.89 ± 0.95 ^b^
S2 ^2^	65.13 ± 1.53 ^c^
S3 ^3^	73.30 ± 0.62 ^c^
S4 ^4^	84.52 ± 0.78 ^a^
S5 ^5^	76.21 ± 1.21 ^b^

^1^ (0.3:1); ^2^ (0.3:2); ^3^ (0.3:3); ^4^ (0.3:4); ^5^ (0.3:5). The ratio of which is gellan gum and soy protein isolate. Results are average values ± standard deviation of triplicate analyses. Different letters indicate statistically significant differences (*p* < 0.05).

**Table 2 foods-11-03729-t002:** Values of fitted parameters by different mathematical models.

Model	Fitted Parameters	SIF ^1^	SGF ^2^	Whole
Zero-order	r^2^	0.9799	0.9720	0.9728
	k_0_	0.0791	0.1730	0.1711
Peppas	r^2^	0.9823	0.9721	0.9772
	k	0.0524	0.1607	0.0801
	n	1.0901	1.0125	1.1292

^1^ Indicates simulated gastric fluid digestive solution and ^2^ indicates simulated intestinal fluid digestive solution.

**Table 3 foods-11-03729-t003:** Inhibition of α-glucosidase and α-amylase activities by acarbose, kidney tea saponin, and saponin-loaded gels.

	α-Amylase			α-Glucosidase		
	Regression Equation	R^2^	IC_50_ (mg/mL)	Regression Equation	R^2^	IC_50_ (mg/mL)
Acarbose	y = 29.219x + 35.467	0.9980	0.4974	y = 48.662x + 4.7834	0.9987	0.9293
Saponin	y = 19.964x + 20.068	0.9983	1.4995	y = 10.016x + 29.312	0.9996	2.0565
GG/SPI-saponin ^1^	y = 10.025x + 15.576	0.9966	1.4317	y = 9.9969x + 25.213	0.9999	2.4790

Note: ^1^ indicates gellan gum-soybean isolate protein composite gel loaded with saponin. “x” represents the different inhibitor concentrations, “y” represents the inhibition rate.

## Data Availability

The data presented in this study are available on request from the corresponding author.

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
