# Peer review of "Hypoglycemic Activity of Self-Assembled Gellan Gum-Soybean Isolate Composite Hydrogel-Embedded Active Substance-Saponin"

_foods, 2022, doi:10.3390/foods11223729_

Round 1

Reviewer 1 Report

Manuscript: Hypoglycemic activity of self-assembled gellan gum-soybean isolate composite hydrogel-embedded active substance-saponin

Journal: Foods

This paper considers an interesting application of new biomaterials and their properties for medicine potential applications. The authors of the paper were careful in their approach and answered their research questions faithfully but they should be considered to improve more the discussion of their results including some conclusion. In the authors will follow my recommendations and comments, this paper has sufficient relevance and importance. This reviewer would like to recommend a minor revision and minor suggestions are as below:

Page 1-Introduction

Line 44: correct “as the temperature decreases. gel formation” to as the temperature decreases gel formation.

Page 4-Materials and Methods

Line 182: Please, check the space between letters.

Page 5

Line 201: Please, check the space between letters.

Line 212: Please, correct “Statistic alanalysis” to Statistic analysis

Results

Page 6

The sub-title of Fig. 1, please clarify the text, it is unclear.

Page 7

Lines 265, 268, 269: Please, check the space between letters.

Page 11

Line 366: Please, check the space between letters.

Discussion

Line 435: This reviewer carefully recommends checking very well that line, there is some reference for confirm this sentence?

This reviewer carefully suggests reviewing and adding at the same time more discussion including the following parts in this manuscript (or accordance to the journal format - could be included as results and discussion in one part section in the manuscript):

- Encapsulation efficiency

- Texture analysis of gel

- Rheological analysis

- Structural characteristics of composite gel

- Scanning electron microscope (SEM)

- In vitro release mechanism

- In vitro hypoglycemic activity

The conclusion must be included in this manuscript.

Author Response

Dear professors,

We have carefully read the comments suggested for our manuscript and we really appreciate them. It has improved our work considerably. Based on these comments and suggestions, we have made careful modifications. All changes made to the text are highlighted in red. We hope the revised manuscript will meet the journal’s standard.

Sincerely,

Tao Wu

State Key Laboratory of Nutrition and Safety,

Tianjin University of Science and Technology

Responds to the reviewer’s comments:

Reviewer 1

Page 1-Introduction

Line 44: correct “as the temperature decreases. gel formation” to as the temperature decreases gel formation.

Response: Thanks for your comments.We have made revision in the introduction. For the corresponding correction, please see line 59.

Page 4-Materials and Methods

Line 182: Please, check the space between letters.

Response: Thank you for comments. We have checked the spaces in line 182 and corrected them. For the corresponding correction, please see line 198.

Page 5

Line 201: Please, check the space between letters.

Response: Thank you for comments. We have checked the spaces in line 201 and corrected them. For the corresponding correction, please see line 217.

Line 212: Please, correct “Statistic alanalysis” to Statistic analysis

Response: Thank you for comments. We have modified the word “analysis”. For the corresponding correction, please see line 229.

Results

Page 6

The sub-title of Fig. 1, please clarify the text, it is unclear.

Response: Thank you for comments. We have marked clearly in the text. For the corresponding correction, please see lines 120-122.

Page 7

Lines 265, 268, 269: Please, check the space between letters.

Response: Thank you for comments. We have checked the spaces in lines 265, 268, 269 and corrected them. For the corresponding correction, please see line 279. As a part of the content was added in front, so the location of the diagram was modified, please check.

Page 11

Line 366: Please, check the space between letters.

Response: Thank you for comments. We have checked the spaces in line 366 and corrected them. For the corresponding correction, please see line 392.

Discussion

Line 435: This reviewer carefully recommends checking very well that line, there is some reference for confirm this sentence?

Response: Thank you for comments. We have included the appropriate references to substantiate our claims. The newly added reference is [57]. For the corresponding correction, please see line 449.

This reviewer carefully suggests reviewing and adding at the same time more discussion including the following parts in this manuscript (or accordance to the journal format - could be included as results and discussion in one part section in the manuscript):

- Encapsulation efficiency

- Texture analysis of gel

- Rheological analysis

- Structural characteristics of composite gel

- Scanning electron microscope (SEM)

- In vitro release mechanism

- In vitro hypoglycemic activity

The conclusion must be included in this manuscript.

Response: Thank you for comments. We have made the additions, which are in lines 455-479. The details are as follows.

In this study, food-grade proteins and natural microbial polysaccharides were used to develop a GG/SPI-Saponin composite hydrogel. The GG/SPI-Saponin system had the best encapsulation efficiency of 84.52±0.78% at S4; the variation of SPI concentration affected the hardness of the hydrogels, with the concentration reaching the maximum at 3%. The dynamic rheological analysis shows that gels are formed for S0-S5 and the energy storage modulus (G') increases with increasing SPI concentration; the relationship between gel and temperature is investigated and it is found that S0-S5 gels have a certain temperature dependence and the gel-sol transition temperature is different for gels with different SPI concentrations. Microstructure analysis in the first Fourier infrared analysis, the main force between the gels to carry out bonding is hydrogen bonding; then scanning electron microscopy structure shows that compared with S0, the addition of SPI makes the pores of the composite gel gradually become uniform and dense, and the hydrogel at S4 is the most neat, smooth, dense and homogeneous. This is also consistent with the previous study of encapsulation efficiency. The hydrogel has a self-assembled system with a three-dimensional porous network structure that protects the stability of saponin in gastric digestion and improves the release ratio of saponin in small intestine digestion. In vitro digestion experiments showed that saponins were retained in the stomach and subsequently released in the small intestine. The enhanced electrostatic and hydrogen bonding interactions as well as the tight three-dimensional network structure were the main mechanisms of the GG/SPI-Saponin complex gel to regulate the release. It is noteworthy that the encapsulated saponins remained with significant hypoglycemic ability. Thus, our study presents an effective and safe method to protect saponins from degradation in the stomach and deliver them to the intestine, and GG/SPI holds great promise as a potential delivery system to improve the stability and bioavailability of functional agents.

Reviewer 2 Report

1. In the Introduction section authors should discuss possible negative effects of saponins.

2. The triterpene saponines and their hypoglycemic effects are described by the only reference to the work of 2007 year. Please add more discussion and new references.

3. It is not clear what sapoin is described as renal/kidney tea saponin. Please add description, chemical structure, describe how is it used in nutrition or medicine. Is it one single saponinOr mixtureOr some kind of extract?

4. Beneficial effects of soybean protein are also presented by pretty old reference from 2010, what also needs some revision.

Minor revisions  

  1. Lines 224, 226 What means abbreviation «VC»?
  1. Line 227 What means abbreviation «EE»?
  1. Kidney tea / renal tea — please keep the terms consistent.

Author Response

Dear professors,

We have carefully read the comments suggested for our manuscript and we really appreciate them. It has improved our work considerably. Based on these comments and suggestions, we have made careful modifications. All changes made to the text are highlighted in red. We hope the revised manuscript will meet the journal’s standard.

Sincerely,

Tao Wu

State Key Laboratory of Nutrition and Safety,

Tianjin University of Science and Technology

Reviewer 2

  1. In the Introduction section authors should discuss possible negative effects of saponins.

Response: Thank you for comments. We have added new content and references, which are in lines 50-54. New references added are [10]-[11]. The specific modifications, as follows.

Whereas saponins are hemolytic in nature, saponins can form complexes with sterols on the erythrocyte membrane, leading to increased permeability and subsequent loss of hemoglobin [10], and also induce intestinal inflammation and impair intestinal barrier function [11]. Therefore the scope of its application is limited.

  1. The triterpene saponines and their hypoglycemic effects are described by the only reference to the work of 2007 year. Please add more discussion and new references.

Response: Thank you for comments. We have made the additions, which are in lines 43-50. New references added are [7]-[9]. The specific modifications, as follows.

The potential hypoglycemic activity of triterpene saponins is partly due to their ability to inhibit various enzymes in the small intestine, including α-glucosidase, protein tyrosine phosphate (PTP1B) [7], and α-amylase [8]. On the other hand, saponins can protect the pancreatic β-cell lineage from high glucose damage [9]. In diabetes, hyperglycemia generates reactive oxygen species (ROS), which in turn lead to lipid peroxidation and membrane damage, and these free radicals play an important role in the development of secondary complications of diabetes (renal, ocular, vascular and neurological damage) [9].

  1. It is not clear what sapoin is described as renal/kidney tea saponin. Please add description, chemical structure, describe how is it used in nutrition or medicine. Is it one single saponin? Or mixture? Or some kind of extract?

Response: Thank you for comments. We have made the additions, which are in lines 32-41. New references added are [2]-[6]. The specific modifications, as follows.

Kidney tea, also known as cat's whiskers and buddleia, is a perennial herb of the genus Clerodendranthus in the family Lamiaceae. The chemical components that have been isolated include flavonoids, phenolic acids and terpenoids, among which the terpenoids are mainly diterpenoids and triterpenoids, and triterpene saponins are the main class of saponins, consisting of 30 carbon skeletal triterpene glycosides [2].

Saponins have numerous physical properties including foaming and emulsification [3] and are used to make detergents, foaming agents and emulsions. Their pharmacological effects namely anticancer and inhibition of cytotoxicity [4], antidiabetic activity [5], anti-inflammatory activity and immunomodulatory activity [6].

  1. Beneficial effects of soybean protein are also presented by pretty old reference from 2010, what also needs some revision.

Response: Thank you for comments. We have revised and replaced outdated literature, which are in lines 75-88. New references added are [18]-[26]. The specific modifications, as follows.

Soybean isolate protein (SPI) is a resource-rich, inexpensive, high quality vegetable protein with high nutritional value, cholesterol-lowering, blood pressure-lowering [18], blood sugar-lowering [19] and good gelling and emulsifying effects [20], mainly composed of 7S and 11S [21]. SPI solidifies the protein matrix by de-folding, dissociating and forming aggregates through changes in hydrogen bonding, Van der Waals forces and hydrophobic interactions when heated. SPI hydrogels can encapsulate nutrients in the network structure [22] and are used for targeted delivery of nutrients due to their solubilization as well as release properties [23]. However, when SPI is used as a carrier of nutrients, this deficiency can be well avoided due to the fact that proteins are easily degraded by pepsin when they enter the human digestive tract and reach the stomach, thus releasing a large amount of nutrients in combination with the crystalline gels [24]. Compared to synthetic polymers, protein-based hydrogels are popular in terms of amphiphilicity, biodegradability, biocompatibility [25], high nutritional value, excellent functional properties and low toxicity [26].

Minor revisions  

Lines 224, 226 What means abbreviation «VC»?

Response: Thank you for comments. We have modified VC to vitamin C (VC). For the corresponding correction, please see line 241.

Line 227 What means abbreviation «EE»?

Response: Thank you for comments. We have modified EE to Encapsulation efficiency (EE). For the corresponding correction, please see line 238.

Kidney tea / renal tea — please keep the terms consistent.

Response: Thank you for comments. We have changed the kidney tea in the full text to be consistent and marked in the text, please check.

Round 2

Reviewer 1 Report

Dear Editor,

I cannot see any improvement in the discussion part including a short conclusion in the final part. I strongly recommend to the authors work in this part. 

Author Response

Response: Thanks for your comments. We have revised and improved the article based on your suggestions. We have analyzed and discussed the experiment in detail in Results and Discussions, and added a Conclusions section. We have highlighted the changes in red.

Results and Discussions

3.1 Encapsulation efficiency (Lines 238-252)

Encapsulation efficiency (EE) is an important characteristic of hydrogels, as shown in Table 1, except for S0 (blank), the encapsulation rate of composite gels increased gradually with increasing SPI concentration, S4 reached a maximum of 84.52 ± 0.78% at SPI concentration of 3%. For instance, the encapsulation rates of vitamin C (VC) in liposomes and chitosan nanoparticles were 48.30% [34] and 15.70% [35], respectively. In contrast, the self-assembled hydrogel formed by bovine serum albumin-citrus peel pectin reached 65.3% of VC encapsulation [36]. Spray drying of maltose dextrin resulted in an EE of 73.81% for steviol glycosides, while electrospraying with corn alcohol soluble protein increased the EE to 75.87% [37]. The addition of SPI significantly increased the viscosity of the hydrogel system, and with the increase of SPI content, the embedding rate showed a trend of first increasing and then decreasing, indicating that the viscosity of the composite gel system was too large and also unfavorable for the embedding of saponins, and S4 was considered to develop the most suitable experimental conditions for the development of thermally stable hydrogels of gellan gum and soybean isolate protein.

3.2. Texture analysis of gel (Lines 262-272)

The hydrogel gel strength was influenced by the soybean isolate protein content. As shown in Figure 1, the addition of soybean isolate protein to the gellan gum can improve the gel strength of the composite hydrogel, and the strength of the hydrogel increases with the increase of SPI content. The addition of too much can lead to excessive viscosity of the hydrogel thermal solution making it impossible to mix the encapsulated material uniformly, thus making the hardness decrease[38]. When the amount of soybean isolate protein exceeds the optimal dose, the gel strength of the hydrogel decreases with the increase of protein content, which may be due to the collapse of the three-dimensional network of the hydrogel caused by too much protein filling, making the hardness decrease. S3 composite hydrogel has the highest gel strength, while the gel strength of S4 and S5 composite hydrogels gradually decreased.

3.3. Rheological analysis (Lines 274-286, Lines 290-306)

The dynamic rheological properties of the GG/SPI composite system can reflect the viscoelasticity of the composite system. The energy storage modulus G' reflects the elastic nature of the stored energy in the system and can recover its original shape, and is related to the strength of the junction zone, the number of bonding and the effective chains forming the network structure [39]; G" can represent the viscosity of the sample, which is influenced by the friction, motion and mobility of small molecules [40]. As shown in Figure 2, the energy storage moduli G' and G'' of the hydrogels gradually increased with the increase of soybean isolate protein content, except for S0. The junction zone expanded with increasing biopolymer concentration, indicating that the increase of soybean isolate protein affects the rheological properties of the composite carrier structure. It can also be seen from the figure that the energy storage modulus (G') of all hydrogels is higher than their loss modulus (G") regardless of the SPI concentration, indicating the formation of GG/SPI hydrogels.

As shown in Figure 3, the G' and G" curves of S0 show an increasing trend after the G'-G" intersection at 67.15°C, which is the gel-sol transition temperature (Tsg) of S0. This is mainly due to the fact that the gellan gum shows a double helical structure at low temperature and undergoes irregular curling with the increase of temperature. The temperatures at which G' intersects G" for S1-S5 are at 67.43°C, 64.98°C, 58.62°C, 58.89°C and 74.65°C, respectively. In the curves of G' or G" versus temperature for S0-S5, the energy storage modulus G' gradually decreases with increasing temperature, intersects with G" at the corresponding temperature, and G' is greater than G" before intersection, which indicates that a gel is formed at low temperature, while with the increase of temperature, G' is smaller than G" and a sol is formed, realizing the gel-sol transition. The increase in temperature led to a decrease in G' of the soybean isolate protein suspension, which could be attributed to the loss of stiffness due to the reduction of subcarbon hydrogen bonds and electrostatic interactions in the protein molecules as a result of the initial heating[41]. When the temperature was further increased, the SPI molecules exposed more reactive groups and formed a sparse gel structure. In conclusion, the rheological properties of GG/SPI gels can be controlled by varying the concentration of SPI.

3.4. Structural characteristics of composite gel (Lines 314-331) 

3.4.1. Fourier transform infrared spectroscopy (FTIR)

The structural characteristics and thermal stability of S4 were investigated by infrared spectroscopy and thermogravimetry, respectively, and we used infrared spectroscopy to understand the interactions between gellan gum, soybean isolate and kidney tea saponin. As shown in Figure 4A, the main characteristic absorption peaks of SPI are the C=O stretching vibration absorption peak in the amide I band at 1672.84 cm-1, the N-H bending vibration absorption peak in the amide II band at 1523.83 cm-1, and the absorption peak at 1238.15 cm-1 caused by the C-H stretching vibration and N-H bending vibration in the amide III band, as shown in Figure 4A. The C-H bending vibration peak at 1450.72 cm-1, the absorption peak at 1390.10 cm-1 due to asymmetric deformation of CH3, the C-O stretching vibration peak at 1076.46 cm-1, and the broader peak at 3286.22 cm-1 for the N-H stretching vibration of the protein [42]. The absorption bands of the GG were 3415, 2927, 1635 and 1418 cm-1, corresponding to the O-H stretching vibration, the C-H stretching vibration, the C-O stretching vibration of the carboxylic acid anion and the C-H stretching vibration of the methyl group [43]. The difference between GG/SPI-Saponin and saponin indicates that the O-H absorption band shifts from 3432 cm-1 to 3395 cm-1. Hydrogen bond formation reduces the density of bonding electron clouds, chemical bonding force constants and stretching vibrational absorption shifts to lower wave numbers [44]. Thus, the findings also suggest that the main interaction between the gellan gum and soybean isolate is hydrogen bonding.

3.4.2. Thermogravimetric analysis (TGA) (Lines 333-347)

Thermogravimetric analysis was performed to study the mass loss and degradation temperature of GG, SPI, saponin, GG-SPI and GG/SPI-Saponin to further determine their thermal stability. The lower sample mass loss rate reflects the higher stability of the samples at the same temperature. The thermal degradation of the hydrogels was determined over the temperature range of 30-500°C. As shown in Figure 4B, the thermal degradation of pure saponins was first at 110°C, where a mass loss of 3.99% occurred. A mass loss of 2.26% for the temperature range of 110-175.93°C mainly by physical and chemical evaporation of water [45]. The degradation at 175.93-500°C is the degradation of glycosidic bond breaking and intramolecular hydrogen bonding. The mass loss of GG/SPI-Saponin was distributed between 30-101.34°C (6.68% mass loss), 101.34-196.08°C (2.27% mass loss), 196.08-500°C (59.39% mass loss). The causes of pyrolysis are evaporation of water vapor, breakage of side chains, and degradation of the main chain [46]. The maximum degradation temperature values of GG/SPI-Saponin composite gels were significantly higher compared to pure saponin, indicating an improved thermal stability of saponin.

3.5. Scanning electron microscope (SEM) (Lines 354-368)

As shown in Figure 5A-F are the images of S0-S5 magnified by 100x, respectively. The SEM images of GG/SPI composite hydrogels show that the pure gellan gum hydrogel (S0) exhibited an inhomogeneous and rough structure with a lamellar aggregate honeycomb-like 3D network structure, and the pores of S0 were significantly larger than those of the other hydrogels. The addition of soybean isolate reduced the pore size and increased the denseness of the hydrogels. The gel structure becomes more compact and ordered with very small pores, indicating a higher cross-link density of the polymer molecules[48]. The S1 and S2 composite hydrogels exhibit some denseness, while the pores of the composite gels gradually become neat with increasing SPI concentration, and SPI dominates the GG/SPI gel structure. At S4 the pores are smoothly dense and homogeneous. When its concentration is increased again, the composite hydrogel pores start to become larger and looser, which is consistent with the study of encapsulation efficiency mentioned previously. It also indicates that the prepared composite hydrogels can regulate their pore size by controlling the concentration of SPI and have great potential for the encapsulation and delivery of water-soluble nutrients.

3.6. Confocal laser scanning microscope (CLSM) (Lines 370-386)

The distribution of polysaccharides in the GG-SPI gel in the protein was observed by laser confocal microscopy, and the results are shown in Figure 6. Green is GG labeled with FITC and red is SPI labeled with Nile Blue, and GG is more uniformly distributed in SPI [41]. The overlapping yellow region consisting of SPI region (red) and GG region (green) indicates the formation of an interconnected structure between these two substances. As can be seen from the figure, the SPI concentration gradually increases and the yellow region shows a trend of increasing and then decreasing, reaching a maximum at S4, indicating that the two substances bind best at this concentration. This may be due to the heating of soybean isolate protein 7S globulin denaturation so that its hydrophilic group attached to the surface, hydrophilic ability to enhance the hydrogen bonding and charge interaction between soybean isolate protein and the anionic gellan gum, so that the two are better combined, which is also consistent with the results of infrared spectroscopy. Yet again, the increase in the content of the gellan gum resulted in the inability to fuse better, indicating that the binding between the two had reached saturation [49]. Yet again, because of the increased SPI content, larger irregular spherical aggregates appear, and the formed spherical aggregates may play a role in enhancing the gel hardness of the composite gels.

3.7. In vitro release mechanism (Lines 400-411)

As shown in Figure 7, the GG-SPI-Saponin composite gel were released at a relatively low percentage during the gastric digestion phase. Only about 9.438% of saponin was released after 2 h of incubation in SGF. It is believed that this phenomenon is due to the rapid dehydration and shrinkage of the protein / polysaccharide composite gel after the addition of polysaccharide in acidic conditions in the stomach, which wraps around saponin and hinders the erosion of the protein by pepsin. When it was transferred to SIF and incubated for 4 h, the release of saponin reached 63.73%, and finally in SCF, the final release of saponin reached 77.74%. Here, the large release of saponin in the intestine was mainly due to ion exchange, and in the intestinal environment, the pH shifted to neutral leading to massive water absorption and swelling of the gel and an increase in pore size, which allowed the rapid release of saponin.

4.Conclusions (Lines 477-495)

The increased concentration of SPI allows the GG/SPI-Saponin composite gel to possess stronger rheological properties compared to GG alone, which could improve the competitiveness of gellan gum as a gelling material. The change of absorption peaks in FTIR spectral analysis also confirmed the GG-SPI interaction, indicating that inter- and intra-molecular interactions of GG and SPI promoted the formation of the composite gel. The microstructure of the composite gel can be manipulated by changing the concentration of SPI. The increase in biopolymer concentration due to the increased level of cross-linking resulted in a dense gel structure and homogeneous pores. Meanwhile, the improved gel structure resulted in better encapsulation efficiency of GG/SPI-Saponin composite gels, which was beneficial for improving the loading effect of the gels. In addition, the GG/SPI composite gel protects the stability of saponin in gastric digestion and improves the release ratio of saponin in small intestine digestion. Notably, the encapsulated saponins still showed excellent hypoglycemic activity in in vitro hypoglycemic assays. Therefore, our study used food-grade proteins and natural microbial polysaccharides to develop gellan gum/soybean isolate protein composite hydrogels, proposing an effective and safe method to protect saponins from degradation in the stomach and deliver them to the intestine, and GG/SPI as a potential delivery system with promising applications in improving the stability and bioavailability of functional agents.

Reviewer 2 Report

 The manuscript has been sufficiently improved.

Author Response

Thanks for your comments.